# MAXMIN Q-LEARNING: CONTROLLING THE ESTIMATION BIAS OF Q-LEARNING

**Qingfeng Lan, Yangchen Pan, Alona Fyshe, Martha White**
Department of Computing Science
University of Alberta
Edmonton, Alberta, Canada
{qlan3,pan6,alona,whitem}@ualberta.ca

## ABSTRACT

Q-learning suffers from overestimation bias, because it approximates the maximum action value using the maximum estimated action value. Algorithms have been proposed to reduce overestimation bias, but we lack an understanding of how bias interacts with performance, and the extent to which existing algorithms mitigate bias. In this paper, we 1) highlight that the effect of overestimation bias on learning efficiency is environment-dependent; 2) propose a generalization of Q-learning, called *Maxmin Q-learning*, which provides a parameter to flexibly control bias; 3) show theoretically that there exists a parameter choice for Maxmin Q-learning that leads to unbiased estimation with a lower approximation variance than Q-learning; and 4) prove the convergence of our algorithm in the tabular case, as well as convergence of several previous Q-learning variants, using a novel Generalized Q-learning framework. We empirically verify that our algorithm better controls estimation bias in toy environments, and that it achieves superior performance on several benchmark problems. [1]

## 1 INTRODUCTION

Q-learning (Watkins, 1989) is one of the most popular reinforcement learning algorithms. One of the reasons for this widespread adoption is the simplicity of the update. On each step, the agent updates its action value estimates towards the observed reward and the estimated value of the maximal action in the next state. This target represents the highest value the agent thinks it could obtain from the current state and action, given the observed reward.

Unfortunately, this simple update rule has been shown to suffer from overestimation bias (Thrun & Schwartz, 1993; van Hasselt, 2010). The agent updates with the maximum over action values might be large because an action's value actually is high, or it can be misleadingly high simply because of the stochasticity or errors in the estimator. With many actions, there is a higher probability that one of the estimates is large simply due to stochasticity and the agent will overestimate the value. This issue is particularly problematic under function approximation, and can significant impede the quality of the learned policy (Thrun & Schwartz, 1993; Szita & Lőrincz, 2008; Strehl et al., 2009) or even lead to failures of Q-learning (Thrun & Schwartz, 1993). More recently, experiments across several domains suggest that this overestimation problem is common (Hado van Hasselt et al., 2016).

Double Q-learning (van Hasselt, 2010) is introduced to instead ensure *under*estimation bias. The idea is to maintain two unbiased independent estimators of the action values. The expected action value of estimator one is selected for the maximal action from estimator two, which is guaranteed not to overestimate the true maximum action value. Double DQN (Hado van Hasselt et al., 2016), the extension of this idea to Q-learning with neural networks, has been shown to significantly improve performance over Q-learning. However, this is not a complete answer to this problem, because trading overestimation bias for underestimation bias is not always desirable, as we show in our experiments.

---

[1]Code is available at https://github.com/qlan3/Explorer

Several other methods have been introduced to reduce overestimation bias, without fully moving towards underestimation. Weighted Double Q-learning (Zhang et al., 2017) uses a weighted combination of the Double Q-learning estimate, which likely has underestimation bias, and the Q-learning estimate, which likely has overestimation bias. Bias-corrected Q-Learning (Lee et al., 2013) reduces the overestimation bias through a bias correction term. Ensemble Q-learning and Averaged Q-learning (Anschel et al., 2017) take averages of multiple action values, to both reduce the overestimation bias and the estimation variance. However, with a finite number of action-value functions, the average operation in these two algorithms will never completely remove the overestimation bias, as the average of several overestimation biases is always positive. Further, these strategies do not guide how strongly we should correct for overestimation bias, nor how to determine—or control—the level of bias.

The overestimation bias also appears in the actor-critic setting (Fujimoto et al., 2018; Haarnoja et al., 2018). For example, Fujimoto et al. (2018) propose the Twin Delayed Deep Deterministic policy gradient algorithm (TD3) which reduces the overestimation bias by taking the minimum value between two critics. However, they do not provide a rigorous theoretical analysis for the effect of applying the minimum operator. There is also no theoretical guide for choosing the number of estimators such that the overestimation bias can be reduced to 0.

In this paper, we study the effects of overestimation and underestimation bias on learning performance, and use them to motivate a generalization of Q-learning called Maxmin Q-learning. Maxmin Q-learning directly mitigates the overestimation bias by using a minimization over multiple action-value estimates. Moreover, it is able to control the estimation bias varying from positive to negative which helps improve learning efficiency as we will show in next sections. We prove that, theoretically, with an appropriate number of action-value estimators, we are able to acquire an unbiased estimator with a lower approximation variance than Q-learning. We empirically verify our claims on several benchmarks. We study the convergence properties of our algorithm within a novel Generalized Q-learning framework, which is suitable for studying several of the recently proposed Q-learning variants. We also combine deep neural networks with Maxmin Q-learning (Maxmin DQN) and demonstrate its effectiveness in several benchmark domains.

## 2 PROBLEM SETTING

We formalize the problem as a Markov Decision Process (MDP), $(\mathcal{S}, \mathcal{A}, \mathrm{P}, r, \gamma)$, where $\mathcal{S}$ is the state space, $\mathcal{A}$ is the action space, $\mathrm{P} : \mathcal{S} \times \mathcal{A} \times \mathcal{S} \to [0, 1]$ is the transition probabilities, $r : \mathcal{S} \times \mathcal{A} \times \mathcal{S} \to \mathbb{R}$ is the reward mapping, and $\gamma \in [0, 1]$ is the discount factor. At each time step $t$, the agent observes a state $S_t \in \mathcal{S}$ and takes an action $A_t \in \mathcal{A}$ and then transitions to a new state $S_{t+1} \in \mathcal{S}$ according to the transition probabilities P and receives a scalar reward $R_{t+1} = r(S_t, A_t, S_{t+1}) \in \mathbb{R}$. The goal of the agent is to find a policy $\pi : \mathcal{S} \times \mathcal{A} \to [0, 1]$ that maximizes the expected return starting from some initial state.

Q-learning is an off-policy algorithm which attempts to learn the state-action values $Q : \mathcal{S} \times \mathcal{A} \to \mathbb{R}$ for the optimal policy. It tries to solve for

$$Q^*(s, a) = \mathbb{E}\Big[R_{t+1} + \max_{a' \in \mathcal{A}} Q^*(S_{t+1}, a') \; \Big| \; S_t = s, A_t = a\Big]$$

The optimal policy is to act greedily with respect to these action values: from each $s$ select $a$ from $\arg\max_{a \in \mathcal{A}} Q^*(s, a)$. The update rule for an approximation $Q$ for a sampled transition $s_t, a_t, r_{t+1}, s_{t+1}$ is:

$$Q(s_t, a_t) \leftarrow Q(s_t, a_t) + \alpha(Y_t^Q - Q(s_t, a_t)) \qquad \text{for } Y_t^Q \stackrel{\text{def}}{=} r_{t+1} + \gamma \max_{a' \in \mathcal{A}} Q(s_{t+1}, a') \qquad (1)$$

where $\alpha$ is the step-size. The transition can be generated off-policy, from any behaviour that sufficiently covers the state space. This algorithm is known to converge in the tabular setting (Tsitsiklis, 1994), with some limited results for the function approximation setting (Melo & Ribeiro, 2007).

## 3 UNDERSTANDING WHEN OVERESTIMATION BIAS HELPS AND HURTS

In this section, we briefly discuss the estimation bias issue, and empirically show that both overestimation and underestimation bias may improve learning performance, depending on the

environment. This motivates our Maxmin Q-learning algorithm described in the next section, which allows us to flexibly control the estimation bias and reduce the estimation variance.

The overestimation bias occurs since the target $\max_{a' \in \mathcal{A}} Q(s_{t+1}, a')$ is used in the Q-learning update. Because $Q$ is an approximation, it is probable that the approximation is higher than the true value for one or more of the actions. The maximum over these estimators, then, is likely to be skewed towards an overestimate. For example, even unbiased estimates $Q(s_{t+1}, a')$ for all $a'$, will vary due to stochasticity. $Q(s_{t+1}, a') = Q^*(s_{t+1}, a') + e_{a'}$, and for some actions, $e_{a'}$ will be positive. As a result, $\mathbb{E}[\max_{a' \in \mathcal{A}} Q(s_{t+1}, a')] \geq \max_{a' \in \mathcal{A}} \mathbb{E}[Q(s_{t+1}, a')] = \max_{a' \in \mathcal{A}} Q^*(s_{t+1}, a')$.

This overestimation bias, however, may not always be detrimental. And, further, in some cases, erring towards an underestimation bias can be harmful. Overestimation bias can help encourage exploration for overestimated actions, whereas underestimation bias might discourage exploration. In particular, we expect more overestimation bias in highly stochastic areas of the world; if those highly stochastic areas correspond to high-value regions, then encouraging exploration there might be beneficial. An underestimation bias might actually prevent an agent from learning that a region is high-value. Alternatively, if highly stochastic areas also have low values, overestimation bias might cause an agent to over-explore a low-value region.

We show this effect in the simple MDP, shown in Figure 1. The MDP for state $A$ has only two actions: Left and Right. It has a deterministic neutral reward for both the Left action and the Right action. The Left action transitions to state $B$ where there are eight actions transitions to a terminate state with a highly stochastic reward. The mean of this stochastic reward is $\mu$. By selecting $\mu > 0$, the stochastic region becomes high-value, and we expect overestimation bias to help and underestimation bias to hurt. By selecting $\mu < 0$, the stochastic region becomes low-value, and we expect overestimation bias to hurt and underestimation bias to help.

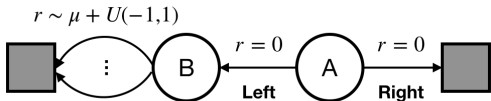

Figure 1: A simple episodic MDP, adapted from Figure 6.5 in Sutton & Barto (2018) which is used to highlight the difference between Double Q-learning and Q-learning. This MDP has two non-terminal states $A$ and $B$. Every episode starts from $A$ which has two actions: Left and Right. The Right action transitions to a terminal state with reward 0. The Left action transitions to state $B$ with reward 0. From state $B$, there are 8 actions that all transition to a terminal state with a reward $\mu + \xi$, where $\xi$ is drawn from a uniform distribution $U(-1, 1)$. When $\mu > 0$, the optimal action in state $A$ is Left; when $\mu < 0$, it is Right.

We test Q-learning, Double Q-learning and our new algorithm Maxmin Q-learning in this environment. Maxmin Q-learning (described fully in the next section) uses $N$ estimates of the action values in the targets. For $N = 1$, it corresponds to Q-learning; otherwise, it progresses from overestimation bias at $N = 1$ towards underestimation bias with increasing $N$. In the experiment, we used a discount factor $\gamma = 1$; a replay buffer with size 100; an $\epsilon$-greedy behaviour with $\epsilon = 0.1$; tabular action-values, initialized with a Gaussian distribution $\mathcal{N}(0, 0.01)$; and a step-size of 0.01 for all algorithms.

The results in Figure 2 verify our hypotheses for when overestimation and underestimation bias help and hurt. Double Q-learning underestimates too much for $\mu = +1$, and converges to a suboptimal policy. Q-learning learns the optimal policy the fastest, though for all values of $N = 2, 4, 6, 8$, Maxmin Q-learning does progress towards the optimal policy. All methods get to the optimal policy for $\mu = -1$, but now Double Q-learning reaches the optimal policy the fastest, and followed by Maxmin Q-learning with larger $N$.

## 4 MAXMIN Q-LEARNING

In this section, we develop Maxmin Q-learning, a simple generalization of Q-learning designed to control the estimation bias, as well as reduce the estimation variance of action values. The idea is

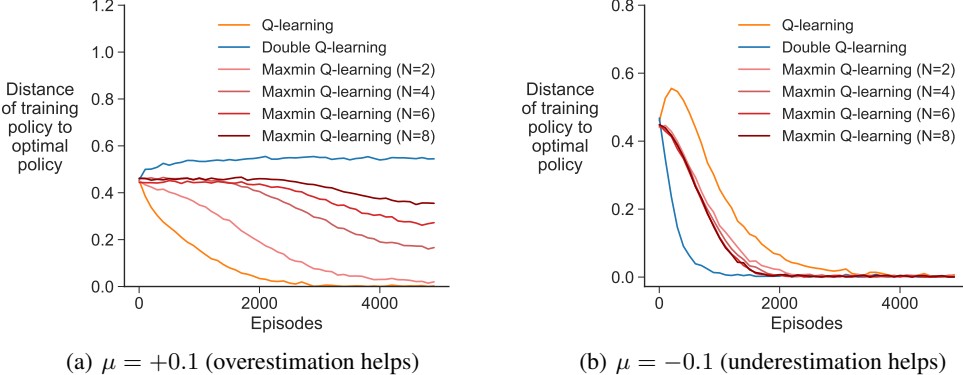

(a) $\mu = +0.1$ (overestimation helps)          (b) $\mu = -0.1$ (underestimation helps)

Figure 2: Comparison of three algorithms using the simple MDP in Figure 1 with different values of $\mu$, and thus different expected rewards. For $\mu = +0.1$, shown in (a), the optimal $\epsilon$-greedy policy is to take the Left action with $95\%$ probability. For $\mu = -0.1$, shown in in (b), the optimal policy is to take the Left action with $5\%$ probability. The reported distance is the absolute difference between the probability of taking the Left action under the learned policy compared to the optimal $\epsilon$-greedy policy. All results were averaged over $5,000$ runs.

to maintain $N$ estimates of the action values, $Q^i$, and use the minimum of these estimates in the Q-learning target: $\max_{a'} \min_{i \in \{1,\dots,N\}} Q^i(s', a')$. For $N = 1$, the update is simply Q-learning, and so likely has overestimation bias. As $N$ increase, the overestimation decreases; for some $N > 1$, this maxmin estimator switches from an overestimate, in expectation, to an underestimate. We characterize the relationship between $N$ and the expected estimation bias below in Theorem 1. Note that Maxmin Q-learning uses a different mechanism to reduce overestimation bias than Double Q-learning; Maxmin Q-learning with $N = 2$ is not Double Q-learning.

The full algorithm is summarized in Algorithm 1, and is a simple modification of Q-learning with experience replay. We use random subsamples of the observed data for each of the $N$ estimators, to make them nearly independent. To do this training online, we keep a replay buffer. On each step, a random estimator $i$ is chosen and updated using a mini-batch from the buffer. Multiple such updates can be performed on each step, just like in experience replay, meaning multiple estimators can be updated per step using different random mini-batches. In our experiments, to better match DQN, we simply do one update per step. Finally, it is also straightforward to incorporate target networks to get Maxmin DQN, by maintaining a target network for each estimator.

We now characterize the relation between the number of action-value functions used in Maxmin Q-learning and the estimation bias of action values. For compactness, we write $Q^i_{sa}$ instead of $Q^i(s, a)$. Each $Q^i_{sa}$ has random approximation error $e^i_{sa}$

$$Q^i_{sa} = Q^*_{sa} + e^i_{sa}.$$

We assume that $e^i_{sa}$ is a uniform random variable $U(-\tau, \tau)$ for some $\tau > 0$. The uniform random assumption was used by Thrun & Schwartz (1993) to demonstrate bias in Q-learning, and reflects that non-negligible positive and negative $e^i_{sa}$ are possible. Notice that for $N$ estimators with $n_{sa}$ samples, the $\tau$ will be proportional to some function of $n_{sa}/N$, because the data will be shared amongst the $N$ estimators. For the general theorem, we use a generic $\tau$, and in the following corollary provide a specific form for $\tau$ in terms of $N$ and $n_{sa}$.

Recall that $M$ is the number of actions applicable at state $s'$. Define the estimation bias $Z_{MN}$ for transition $s, a, r, s'$ to be

$$Z_{MN} \stackrel{\text{def}}{=} (r + \gamma \max_{a'} Q^{min}_{s'a'}) - (r + \gamma \max_{a'} Q^*_{s'a'})$$
$$= \gamma (\max_{a'} Q^{min}_{s'a'} - \max_{a'} Q^*_{s'a'})$$

---

**Algorithm 1:** Maxmin Q-learning

---

**Input:** step-size $\alpha$, exploration parameter $\epsilon > 0$, number of action-value functions $N$
Initialize $N$ action-value functions $\{Q^1, \ldots, Q^N\}$ randomly
Initialize empty replay buffer $D$
Observe initial state $s$
**while** *Agent is interacting with the Environment* **do**
    $Q^{min}(s,a) \leftarrow \min_{k \in \{1,\ldots,N\}} Q^k(s,a), \forall a \in \mathcal{A}$
    Choose action $a$ by $\epsilon$-greedy based on $Q^{min}$
    Take action $a$, observe $r$, $s'$
    Store transition $(s, a, r, s')$ in $D$
    Select a subset $S$ from $\{1, \ldots, N\}$   (e.g., randomly select one $i$ to update)
    **for** $i \in S$ **do**
        Sample random mini-batch of transitions $(s_D, a_D, r_D, s'_D)$ from $D$
        Get update target: $Y^{MQ} \leftarrow r_D + \gamma \max_{a' \in A} Q^{min}(s'_D, a')$
        Update action-value $Q^i$: $Q^i(s_D, a_D) \leftarrow Q^i(s_D, a_D) + \alpha[Y^{MQ} - Q^i(s_D, a_D)]$
    **end**
    $s \leftarrow s'$
**end**

---

where

$$Q^{min}_{sa} \overset{\text{def}}{=} \min_{i \in \{1,\ldots,N\}} Q^i_{sa} = Q^*_{sa} + \min_{i \in \{1,\ldots,N\}} e^i_{sa}$$

We now show how the expected estimation bias $E[Z_{MN}]$ and the variance of $Q^{min}_{sa}$ are related to the number of action-value functions $N$ in Maxmin Q-learning.

**Theorem 1** *Under the conditions stated above,*

*(i) the expected estimation bias is*

$$E[Z_{MN}] = \gamma\tau[1 - 2t_{MN}] \qquad where \; t_{MN} = \frac{M(M-1)\cdots 1}{(M + \frac{1}{N})(M - 1 + \frac{1}{N})\cdots(1 + \frac{1}{N})}.$$

*(ii)* $E[Z_{MN}]$ *decreases as $N$ increases:* $E[Z_{M,N=1}] = \gamma\tau\frac{M-1}{M+1}$ *and* $E[Z_{M,N\to\infty}] = -\gamma\tau$.

$$Var[Q^{min}_{sa}] = \frac{4N\tau^2}{(N+1)^2(N+2)}.$$

$Var[Q^{min}_{sa}]$ *decreases as $N$ increases:* $Var[Q^{min}_{sa}] = \frac{\tau^2}{3}$ *for N=1 and* $Var[Q^{min}_{sa}] = 0$ *for $N \to \infty$.*

Theorem 1 is a generalization of the first lemma in Thrun & Schwartz (1993); we provide the proof in Appendix A as well as a visualization of the expected bias for varying $M$ and $N$. This theorem shows that the average estimation bias $E[Z_{MN}]$, decreases as $N$ increases. Thus, we can control the bias by changing the number of estimators in Maxmin Q-learning. Specifically, the average estimation bias can be reduced from positive to negative as $N$ increases. Notice that $E[Z_{MN}] = 0$ when $t_{MN} = \frac{1}{2}$. This suggests that by choosing $N$ such that $t_{MN} \approx \frac{1}{2}$, we can reduce the bias to near 0.

Furthermore, $Var[Q^{min}_{sa}]$ decreases as $N$ increases. This indicates that we can control the estimation variance of target action value through $N$. We show just this in the following Corollary. The subtlety is that with increasing $N$, each estimator will receive less data. The fair comparison is to compare the variance of a single estimator that uses all of the data, as compared to the maxmin estimator which shares the samples across $N$ estimators. We show that there is an $N$ such that the variance is lower, which arises largely due to the fact that the variance of each estimator decreases linearly in $n$, but the $\tau$ parameter for each estimator only decreases at a square root rate in the number of samples.

**Corollary 1** *Assuming the $n_{sa}$ samples are evenly allocated amongst the $N$ estimators, then $\tau = \sqrt{3\sigma^2 N/n_{sa}}$ where $\sigma^2$ is the variance of samples for $(s, a)$ and, for $Q_{sa}$ the estimator that uses all*

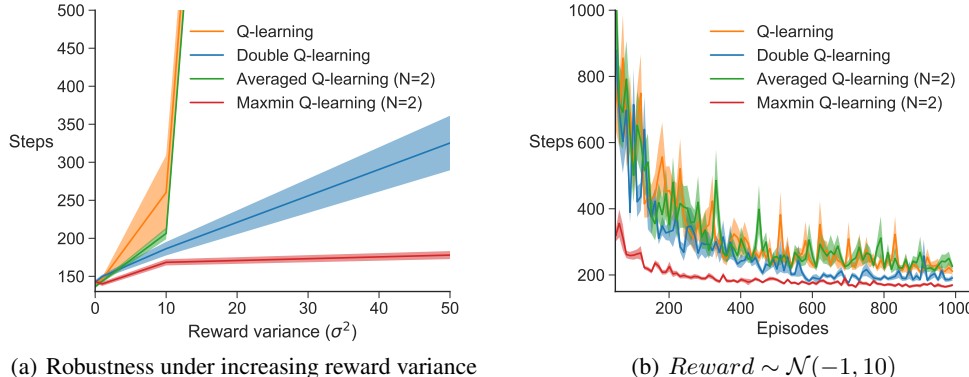

(a) Robustness under increasing reward variance

(b) $Reward \sim \mathcal{N}(-1, 10)$

Figure 3: Comparison of four algorithms on Mountain Car under different reward variances. The lines in $(a)$ show the average number of steps taken in the last episode with one standard error. The lines in $(b)$ show the number of steps to reach the goal position during training when the reward variance $\sigma^2 = 10$. All results were averaged across 100 runs, with standard errors. Additional experiments with further elevated $\sigma^2$ can be found in Appendix C.2.

$n_{sa}$ samples for a single estimate,

$$Var[Q_{sa}^{min}] = \frac{12N^2}{(N+1)^2(N+2)} Var[Q_{sa}].$$

Under this uniform random noise assumption, for $N \geq 8$, $Var[Q_{sa}^{min}] < Var[Q_{sa}]$.

## 5 EXPERIMENTS

In this section, we first investigate robustness to reward variance, in a simple environment (Mountain Car) in which we can perform more exhaustive experiments. Then, we investigate performance in seven benchmark environments.

**Robustness under increasing reward variance in Mountain Car** Mountain Car (Sutton & Barto, 2018) is a classic testbed in Reinforcement Learning, where the agent receives a reward of $-1$ per step with $\gamma = 1$, until the car reaches the goal position and the episode ends. In our experiment, we modify the rewards to be stochastic with the same mean value: the reward signal is sampled from a Gaussian distribution $\mathcal{N}(-1, \sigma^2)$ on each time step. An agent should learn to reach the goal position in as few steps as possible.

The experimental setup is as follows. We trained each algorithm with $1,000$ episodes. The number of steps to reach the goal position in the last training episode was used as the performance measure. The fewer steps, the better performance. All experimental results were averaged over 100 runs. The key algorithm settings included the function approximator, step-sizes, exploration parameter and replay buffer size. All algorithm used $\epsilon$-greedy with $\epsilon = 0.1$ and a buffer size of 100. For each algorithm, the best step-size was chosen from $\{0.005, 0.01, 0.02, 0.04, 0.08\}$, separately for each reward setting. Tile-coding was used to approximate the action-value function, where we used 8 tilings with each tile covering $1/8$th of the bounded distance in each dimension. For Maxmin Q-learning, we randomly chose one action-value function to update at each step.

As shown in Figure 3, when the reward variance is small, the performance of Q-learning, Double Q-learning, Averaged Q-learning, and Maxmin Q-learning are comparable. However, as the variance increases, Q-learning, Double Q-learning, and Averaged Q-learning became much less stable than Maxmin Q-learning. In fact, when the variance was very high ($\sigma = 50$, see Appendix C.2), Q-learning and Averaged Q-learning failed to reach the goal position in $5,000$ steps, and Double Q-learning produced runs $> 400$ steps, even after many episodes.

**Results on Benchmark Environments** To evaluate Maxmin DQN, we choose seven games from Gym (Brockman et al., 2016), PyGame Learning Environment (PLE) (Tasfi, 2016), and MinAtar (Young & Tian, 2019): Lunarlander, Catcher, Pixelcopter, Asterix, Seaquest, Breakout, and Space Invaders. For games in MinAtar (i.e. Asterix, Seaquest, Breakout, and Space Invaders), we reused the hyper-parameters and settings of neural networks in (Young & Tian, 2019). And the step-size was chosen from $[3 * 10^{-3}, 10^{-3}, 3 * 10^{-4}, 10^{-4}, 3 * 10^{-5}]$. For Lunarlander, Catcher, and Pixelcopter, the neural network was a multi-layer perceptron with hidden layers fixed to $[64, 64]$. The discount factor was $0.99$. The size of the replay buffer was $10,000$. The weights of neural networks were optimized by RMSprop with gradient clip 5. The batch size was 32. The target network was updated every 200 frames. $\epsilon$-greedy was applied as the exploration strategy with $\epsilon$ decreasing linearly from 1.0 to 0.01 in $1,000$ steps. After $1,000$ steps, $\epsilon$ was fixed to 0.01. For Lunarlander, the best step-size was chosen from $[3 * 10^{-3}, 10^{-3}, 3 * 10^{-4}, 10^{-4}, 3 * 10^{-5}]$. For Catcher and Pixelcopter, the best step-size was chosen from $[10^{-3}, 3 * 10^{-4}, 10^{-4}, 3 * 10^{-5}, 10^{-5}]$.

For both Maxmin DQN and Averaged DQN, the number of target networks $N$ was chosen from $[2, 3, 4, 5, 6, 7, 8, 9]$. And we randomly chose one action-value function to update at each step. We first trained each algorithm in a game for certain number of steps. After that, each algorithm was tested by running 100 test episodes with $\epsilon$-greedy where $\epsilon = 0.01$. Results were averaged over 20 runs for each algorithm, with learning curves shown for the best hyper-parameter setting (see Appendix C.3 for the parameter sensitivity curves).

We see from Figure 4 that Maxmin DQN performs as well as or better than other algorithms. In environments where final performance is noticeably better—-Pixelcopter, Lunarlander and Asterix—the initial learning is slower. A possible explanation for this is that the Maxmin agent more extensively explored early on, promoting better final performance. We additionally show on Pixelcopter and Asterix that for smaller $N$, Maxmin DQN learns faster but reaches suboptimal performance—behaving more like Q-learning—and for larger $N$ learns more slowly but reaches better final performance.

## 6 CONVERGENCE ANALYSIS OF MAXMIN Q-LEARNING

In this section, we show Maxmin Q-learning is convergent in the tabular setting. We do so by providing a more general result for what we call Generalized Q-learning: Q-learning where the bootstrap target uses a function $G$ of $N$ action values. The main condition on $G$ is that it maintains relative maximum values, as stated in Assumption 1. We use this more general result to prove Maxmin Q-learning is convergent, and then discuss how it provides convergence results for Q-learning, Ensemble Q-learning, Averaged Q-learning and Historical Best Q-learning as special cases.

Many variants of Q-learning have been proposed, including Double Q-learning (van Hasselt, 2010), Weighted Double Q-learning (Zhang et al., 2017), Ensemble Q-learning (Anschel et al., 2017), Averaged Q-learning (Anschel et al., 2017), and Historical Best Q-learning (Yu et al., 2018). These algorithms differ in their estimate of the one-step bootstrap target. To encompass all variants, the target action-value of Generalized Q-learning $Y^{GQ}$ is defined based on action-value estimates from both dimensions:

$$Y^{GQ} = r + \gamma Q_{s'}^{GQ}(t-1) \qquad (2)$$

where $t$ is the current time step and the action-value function $Q_s^{GQ}(t)$ is a function of $Q_s^1(t - K), \ldots, Q_s^1(t-1), \ldots, Q_s^N(t-K), \ldots, Q_s^N(t-1)$:

$$Q_s^{GQ}(t) = G \begin{pmatrix} Q_s^1(t-K) & \ldots & Q_s^1(t-1) \\ Q_s^2(t-K) & \ldots & Q_s^2(t-1) \\ \vdots & \ddots & \vdots \\ Q_s^N(t-K) & \ldots & Q_s^N(t-1) \end{pmatrix} \qquad (3)$$

For simplicity, the vector $(Q_{sa}^{GQ}(t))_{a \in \mathcal{A}}$ is denoted as $Q_s^{GQ}(t)$, same for $Q_s^i(t)$. The corresponding update rule is

$$Q_{sa}^i(t) \leftarrow Q_{sa}^i(t-1) + \alpha_{sa}^i(t-1)(Y^{GQ} - Q_{sa}^i(t-1)) \qquad (4)$$

For different $G$ functions, Generalized Q-learning reduces to different variants of Q-learning, including Q-learning itself. For example, Generalized Q-learning can be reduced to Q-learning

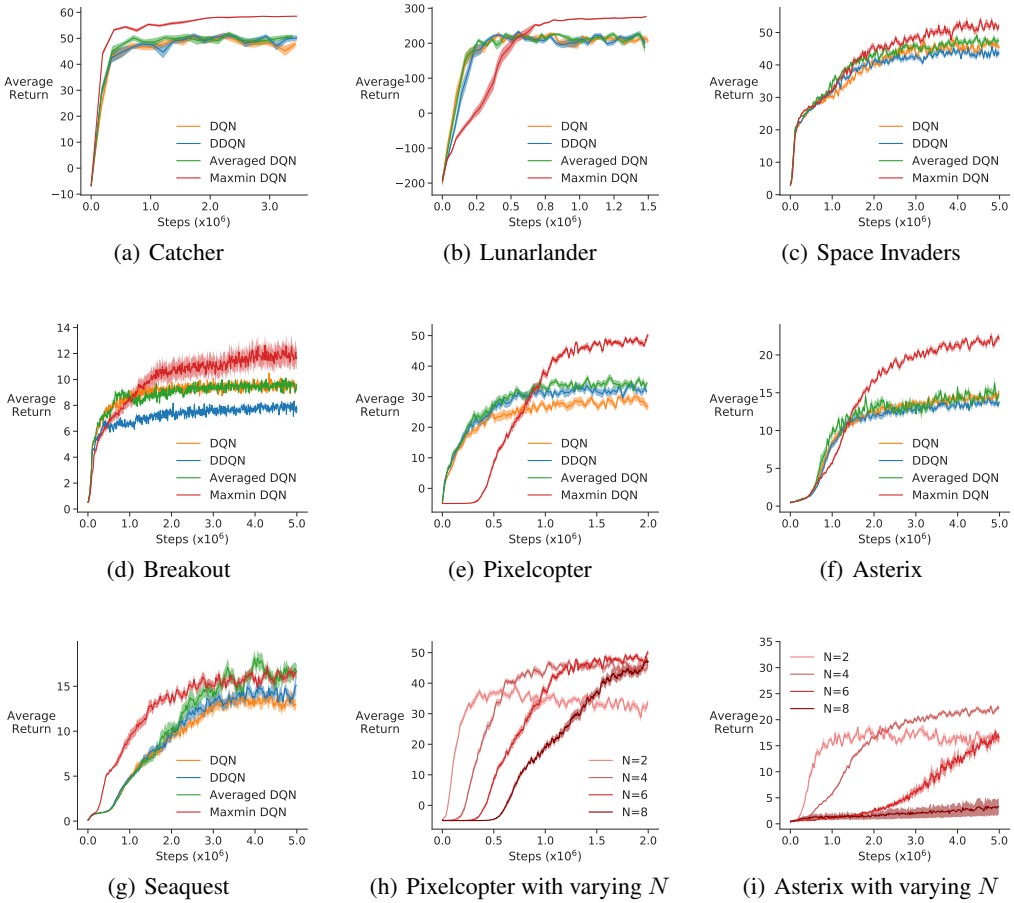

Figure 4: Learning curves on the seven benchmark environments. The depicted return is averaged over the last 100 episodes, and the curves are smoothed using an exponential average, to match previous reported results (Young & Tian, 2019). The results were averaged over 20 runs, with the shaded area representing one standard error. Plots $(h)$ and $(i)$ show the performance of Maxmin DQN on Pixelcopter and Asterix, with different $N$, highlighting that larger $N$ seems to result in slower early learning but better final performance in both environments.

simply by setting $K = 1$, $N = 1$ with $G(Q_s) = \max_{a \in A} Q_{sa}$. Double Q-learning can be specified with $K = 1$, $N = 2$, and $G(Q_s^1, Q_s^2) = Q_{s, \arg\max_{a' \in A} Q_{sa'}^1}^2$.

We first introduce Assumption 1 for function $G$ in Generalized Q-learning, and then state the theorem. The proof can be found in Appendix B.

**Assumption 1 (Conditions on $G$)** *Let $G : \mathbb{R}^{nNK} \mapsto \mathbb{R}$ and $G(Q) = q$ where $Q = (Q_a^{ij}) \in \mathbb{R}^{nNK}$, $a \in \mathcal{A}$ and $|\mathcal{A}| = n$, $i \in \{1, \ldots, N\}, j \in \{0, \ldots, K-1\}$ and $q \in \mathbb{R}$.*

*(i) If $Q_a^{ij} = Q_a^{kl}$, $\forall i, k$, $\forall j, l$, and $\forall a$, then $q = \max_a Q_a^{ij}$.*

*(ii) $\forall Q, Q' \in \mathbb{R}^{nNK}$, $| G(Q) - G(Q') | \le \max_{a,i,j} | Q_a^{ij} - Q'_a^{ij} |$.*

We can verify that Assumption 1 holds for Maxmin Q-learning. Set $K = 1$ and set $N$ to be a positive integer. Let $Q_s = (Q_s^1, \ldots, Q_s^N)$ and define $G^{MQ}(Q_s) = \max_{a \in A} \min_{i \in \{1, \ldots, N\}} Q_{sa}^i$. It is easy to check that part (i) of Assumption 1 is satisfied. Part (ii) is also satisfied because

$$| G(Q_s) - G(Q'_s) | \le | \max_a \min_i Q_{sa}^i - \max_{a'} \min_{i'} Q_{sa'}^{'i'} | \le \max_{a,i} | Q_{sa}^i - Q_{sa}^{'i} | .$$

**Assumption 2 (Conditions on the step-sizes)** *There exists some (deterministic) constant $C$ such that for every $(s, a) \in \mathcal{S} \times \mathcal{A}, i \in \{1, \ldots, N\}$, $0 \le \alpha_{sa}^i(t) \le 1$, and with probability 1,*

$$\sum_{t=0}^{\infty} (\alpha_{sa}^i(t))^2 \le C, \quad \sum_{t=0}^{\infty} \alpha_{sa}^i(t) = \infty$$

**Theorem 2** *Assume a finite MDP $(\mathcal{S}, \mathcal{A}, \mathrm{P}, R)$ and that Assumption 1 and 2 hold. Then the action-value functions in Generalized Q-learning, using the tabular update in Equation (3), will converge to the optimal action-value function with probability 1, in either of the following cases: (i) $\gamma < 1$, or (ii) $\gamma = 1$, $\forall a \in \mathcal{A}, Q_{s_1 a}^i(t = 0) = 0$ where $s_1$ is an absorbing state and all policies are proper.*

As shown above, because the function $G$ for Maxmin Q-learning satisfies Assumption 1, then by Theorem 2 it converges. Next, we apply Theorem 2 to Q-learning and its variants, proving the convergence of these algorithms in the tabular case. For Q-learning, set $K = 1$ and $N = 1$. Let $G^Q(Q_s) = \max_{a \in A} Q_{sa}$. It is straightforward to check that Assumption 1 holds for function $G^Q$. For Ensemble Q-learning, set $K = 1$ and set $N$ to be a positive integer. Let $G^{EQ}((Q_s^1, \ldots, Q_s^N)) = \max_{a \in A} \frac{1}{N} \sum_{i=1}^N Q_{sa}^i$. Easy to check that Assumption 1 is satisfied. For Averaged Q-learning, the proof is similar to Ensemble Q-learning except that $N = 1$ and $K$ is a positive integer. For Historical Best Q-learning, set $N = 1$ and $K$ to be a positive integer. We assume that all auxiliary action-value functions are selected from action-value functions at most $K$ updates ago. Define $G^{HBQ}$ to be the largest action-value among $Q_{sa}(t-1), \ldots, Q_{sa}(t-K)$ for state $s$. Assumption 1 is satisfied and the convergence is guaranteed.

## 7 CONCLUSION

Overestimation bias is a byproduct of Q-learning, stemming from the selection of a maximal value to estimate the expected maximal value. In practice, overestimation bias leads to poor performance in a variety of settings. Though multiple Q-learning variants have been proposed, Maxmin Q-learning is the first solution that allows for a flexible control of bias, allowing for overestimation or underestimation determined by the choice of $N$ and the environment. We showed theoretically that we can decrease the estimation bias and the estimation variance by choosing an appropriate number $N$ of action-value functions. We empirically showed that advantages of Maxmin Q-learning, both on toy problems where we investigated the effect of reward noise and on several benchmark environments. Finally, we introduced a new Generalized Q-learning framework which we used to prove the convergence of Maxmin Q-learning as well as several other Q-learning variants that use $N$ action-value estimates.

### ACKNOWLEDGMENTS

We would like to thank Huizhen Yu and Yi Wan for their valuable feedback and helpful discussion.

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

## A   THE PROOF OF THEOREM 1

We first present Lemma 1 here as a tool to prove Theorem 1. Note that the first three properties in this lemma are well-known results of order statistics (David & Nagaraja, 2004).

**Lemma 1** *Let $X_1, \ldots, X_N$ be $N$ i.i.d. random variables from an absolutely continuous distribution with probability density function(PDF) $f(x)$ and cumulative distribution function (CDF) $F(x)$. Denote $\mu \overset{\text{def}}{=} E[X_i]$ and $\sigma^2 \overset{\text{def}}{=} Var[X_i] < +\infty$. Set $X_{1:N} \overset{\text{def}}{=} min_{i \in \{1,\ldots,N\}} X_i$ and $X_{N:N} \overset{\text{def}}{=} max_{i \in \{1,\ldots,N\}} X_i$. Denote the PDF and CDF of $X_{1:N}$ as $f_{1:N}(x)$ and $F_{1:N}(x)$, respectively. Similarly, denote the PDF and CDF of $X_{N:N}$ as $f_{N:N}(x)$ and $F_{N:N}(x)$, respectively. We then have*

(i) $\mu - \frac{(N-1)\sigma}{\sqrt{2n-1}} \leq E[X_{1:N}] \leq \mu$ and $E[X_{1:N+1}] \leq E[X_{1:N}]$.

(ii) $F_{1:N}(x) = 1 - (1 - F(x))^N$. $f_{1:N}(x) = Nf(x)(1 - F(x))^{N-1}$.

(iii) $F_{N:N}(x) = (F(x))^N$. $f_{N:N}(x) = Nf(x)(F(x))^{N-1}$.

(iv) If $X_1, \ldots, X_N \sim U(-\tau, \tau)$, we have $Var(X_{1:N}) = \frac{4N\tau^2}{(N+1)^2(N+2)}$ and $Var(X_{1:N+1}) < Var(X_{1:N}) \leq Var(X_{1:1}) = \sigma^2$ for any positive integer $N$.

**Proof.**

(i) By the definition of $X_{1:N}$, we have $X_{1:N+1} \leq X_{1:N}$. Thus $E[X_{1:N+1}] \leq E[X_{1:N}]$. Since $E[X_{1:1}] = E[X_1] = \mu$, $E[X_{1:N}] \leq E[X_{1:1}] = \mu$. The proof of $\mu - \frac{(N-1)\sigma}{\sqrt{2N-1}} \leq E[X_{1:N}]$ can be found in (David & Nagaraja, 2004, Chapter 4 Section 4.2).

(ii) We first consider the cdf of $X_{1:N}$. $F_{1:N}(x) := P(X_{1:N} \leq x) = 1 - P(X_{1:N} > x) = 1 - P(X_1 > x, \ldots, X_M > x) = 1 - P(X_1 > x) \cdots P(X_N > x) = 1 - (1 - F(x))^N$. Then the pdf of $X_{1:N}$ is $f_{1:N}(x) := \frac{dF_{1:N}}{dx} = Nf(x)(1 - F(x))^{N-1}$.

(iii) Similar to (ii), we first consider cdf of $X_{N:N}$. $F_{N:N}(x) := P(X_{N:N} \leq x) = P(X_1 \leq x, \ldots, X_N \leq x) = P(X_1 \leq x) \cdots P(X_M \leq x) = (F(x))^N$. Then the pdf of $X_{N:N}$ is $f_{N:N}(x) := \frac{dF_{N:N}}{dx} = Nf(x)(F(x))^{N-1}$.

(iv) Since $X_1, \ldots, X_N \sim Uniform(-\tau, \tau)$, we have $F(x) = \frac{1}{2} + \frac{x}{2\tau}$ and $f(x) = \frac{1}{2\tau}$. $Var(X_{1:N}) = E[X_{1:N}{}^2] - E[X_{1:N}]^2 = 4\tau^2(\frac{2}{(N+1)(N+2)} - \frac{1}{(N+1)^2}) = \frac{4n\tau^2}{(N+1)^2(N+2)}$. It is easy to check that $Var(X_{1:N+1}) < Var(X_{1:N}) \leq Var(X_{1:1}) = \sigma^2$ for any positive integer $N$.

■

Next, we prove Theorem 1.

**Proof.** Let $f(x)$ and $F(x)$ be the cdf and pdf of $e_{sa}$, respectively. Similarly, Let $f_N(x)$ and $F_N(x)$ be the cdf and pdf of $\min_{i \in \{1,\ldots,N\}} e_{sa}^i$. Since $e_{sa}$ is sampled from $Uniform(-\tau, \tau)$, it is easy to get $f(x) = \frac{1}{2\tau}$ and $F(x) = \frac{1}{2} + \frac{x}{2\tau}$. By Lemma 1, we have $f_N(x) = Nf(x)[1 - F(x)]^{N-1} =$

$\frac{N}{2\tau}(\frac{1}{2} - \frac{x}{2\tau})^{N-1}$ and $F_N(x) = 1 - (1 - F(x))^N = 1 - (\frac{1}{2} - \frac{x}{2\tau})^N$. The expectation of $Z_{MN}$ is

$$
\begin{aligned}
E[Z_{MN}] &= \gamma E[(\max_{a'} Q_{s'a'}^{min} - \max_{a'} Q_{s'a'}^*)] \\
&= \gamma E[\max_{a'} \min_{i \in \{1,\dots,N\}} e_{sa'}^i] \\
&= \gamma \int_{-\tau}^{\tau} M x f_N(x) F_N(x)^{M-1} dx \\
&= \gamma \int_{-\tau}^{\tau} MN \frac{x}{2\tau} (\frac{1}{2} - \frac{x}{2\tau})^{N-1} [1 - (\frac{1}{2} - \frac{x}{2\tau})^N]^{M-1} dx \\
&= \gamma \int_{-\tau}^{\tau} x d[1 - (\frac{1}{2} - \frac{x}{2\tau})^N]^M \\
&= \gamma \tau - \gamma \int_{-\tau}^{\tau} [1 - (\frac{1}{2} - \frac{x}{2\tau})^N]^M dx \\
&= \gamma \tau [1 - 2 \int_0^1 (1 - y^N)^M dy] \qquad (y \stackrel{\text{def}}{=} \frac{1}{2} - \frac{x}{2\tau})
\end{aligned}
$$

Let $t_{MN} = \int_0^1 (1 - y^N)^M dy$, so that $E[Z_{MN}] = \gamma \tau [1 - 2t_{MN}]$. Substitute $y$ by $t$ where $t = y^N$, then

$$
\begin{aligned}
t_{MN} &= \frac{1}{N} \int_0^1 t^{\frac{1}{N}-1} (1 - t)^M dt \\
&= \frac{1}{N} B(\frac{1}{N}, M+1) \\
&= \frac{1}{N} \frac{\Gamma(M+1)\Gamma(\frac{1}{N})}{\Gamma(M + \frac{1}{N} + 1)} \\
&= \frac{\Gamma(M+1)\Gamma(1 + \frac{1}{N})}{\Gamma(M + \frac{1}{N} + 1)} \\
&= \frac{M(M-1)\cdots 1}{(M + \frac{1}{N})(M - 1 + \frac{1}{N})\cdots(1 + \frac{1}{N})}
\end{aligned}
$$

Each term in the denominator decreases as $N$ increases, because $1/N$ gets smaller. Therefore, $t_{M,N=1} = \frac{1}{M+1}$ and $t_{M,N\to\infty} = 1$. Using this, we conclude that $E[Z_{MN}]$ decreases as $N$ increases and $E[Z_{M,N=1}] = \gamma\tau \frac{M-1}{M+1}$ and $E[Z_{M,N\to\infty}] = -\gamma\tau$.

By Lemma 1, the variance of $Q_{sa}^{min}$ is

$$
Var[Q_{sa}^{min}] = \frac{4N\tau^2}{(N+1)^2(N+2)}
$$

$Var[Q_{sa}^{min}]$ decreases as $N$ increases. In particular, $Var[Q_{sa}^{min}] = \frac{\tau^2}{3}$ for $N = 1$ and $Var[Q_{sa}^{min}] = 0$ for $N \to \infty$. ∎

The bias-variance trade-off of Maxmin Q-learning is illustrated by the empirical results in Figure 5, which support Theorem 1. For each $M$, $N$ can be selected such that the absolute value of the expected estimation bias is close to 0 according to Theorem 1. As $M$ increases, we can adjust $N$ to reduce both the estimation variance and the estimation bias.

Finally, we prove the result of the Corollary.

**Corollary 1** Assuming the $n_{sa}$ samples are evenly allocated amongst the $N$ estimators, then $\tau = \sqrt{3\sigma^2 N/n_{sa}}$ where $\sigma^2$ is the variance of samples for $(s, a)$ and, for $Q_{sa}$ the estimator that uses all $n_{sa}$ samples for a single estimate,

$$
Var[Q_{sa}^{min}] = \frac{12N^2}{(N+1)^2(N+2)} Var[Q_{sa}].
$$

Under this uniform random noise assumption, for $N \geq 8$, $Var[Q_{sa}^{min}] < Var[Q_{sa}]$.

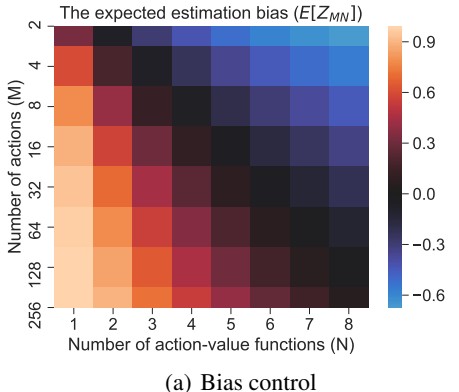
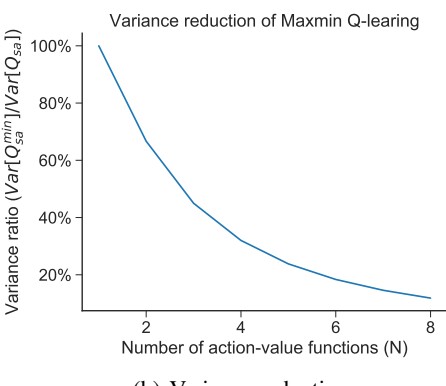

(a) Bias control            (b) Variance reduction

Figure 5: Empirical results of Theorem 1. $M$ is the number of available actions for some state $s$. $N$ is the number of action-value functions in Maxmin Q-learning. In Figure 5 $(a)$, we show a heat map of bias control in Maxmin Q-learning. In Figure 5 $(b)$, we show how the variance ratio of $Q_{sa}^{min}$ and $Q_{sa}$ (i.e. $Var[Q_{sa}^{min}]/Var[Q_{sa}]$) reduces as $N$ increases. For a better comparison, we set $\gamma\tau = 1$.

**Proof.** Because $Q_{sa}^i$ is a sample mean, its variance is $\sigma^2 N/n_{sa}$ where $\sigma^2$ is the variance of samples for $(s, a)$ and its mean is $Q_{sa}^*$ (because it is an unbiased sample average). Consequently, $e_{sa}$ has mean zero and variance $\sigma^2 N/n_{sa}$. Because $e_{sa}$ is a uniform random variable which has variance $\frac{1}{3}\tau^2$, we know that $\tau = \sqrt{3\sigma^2 N/n_{sa}}$. Plugging this value into the variance formula in Theorem 1, we get that

$$Var[Q_{sa}^{min}] = \frac{4N\tau^2}{(N+1)^2(N+2)}$$
$$= \frac{12N^2\sigma^2/n_{sa}}{(N+1)^2(N+2)}$$
$$= \frac{12N^2}{(N+1)^2(N+2)}Var[Q_{sa}]$$

because $Var[Q_{sa}] = \sigma^2/n_{sa}$ for the sample average $Q_{sa}$ that uses all the samples for one estimator. Easy to verify that for $N \geq 8$, $Var[Q_{sa}^{min}] < Var[Q_{sa}]$.

∎

## B THE CONVERGENCE PROOF OF GENERALIZED Q-LEARNING

The convergence proof of Generalized Q-learning is based on Tsitsiklis (1994). The key steps to use this result for Generalized Q-learning include showing that the operator is a contraction and verifying the noise conditions. We first show these two steps in Lemma 2 and Lemma 3. We then use these lemmas to make the standard argument for convergence.

### B.1 PROBLEM SETTING FOR GENERALIZED Q-LEARNING

Consider a Markov decision problem defined on a finite state space $\mathcal{S}$. For every state $s \in \mathcal{S}$, there is a finite set $\mathcal{A}$ of possible actions for state $s$ and a set of non-negative scalars $p_{ss'}(a)$, $a \in \mathcal{A}$, $s' \in \mathcal{S}$, such that $\sum_{j \in S} p_{ss'}(a) = 1$ for all $a \in \mathcal{A}$. The scalar $p_{ss'}(a)$ is interpreted as the probability of a transition to $s'$, given that the current state is $s$ and action $a$ is applied. Furthermore, for every state $s$ and action $a$, there is a random variable $r_{sa}$ which represents the reward if action $a$ is applied at state $s$. We assume that the variance of $r_{sa}$ is finite for every $s$ and $a \in \mathcal{A}$.

A stationary policy is a function $\pi$ defined on $\mathcal{S}$ such that $\pi(s) \in \mathcal{A}$ for all $s \in \mathcal{S}$. Given a stationary policy, we obtain a discrete-time Markov chain $f^\pi(t)$ with transition probabilities

$$\Pr(f^\pi(t+1) = s'|f^\pi(t) = s) = p_{ss'}(\pi(s)) \tag{5}$$

Let $\gamma \in [0, 1]$ be a discount factor. For any stationary policy $\pi$ and initial state $s$, the state value $V_s^\pi$ is defined by

$$V_s^\pi = \lim_{T \to \infty} E[\sum_{t=0}^{T} \gamma^t r_{f^\pi(t), \pi(f^\pi(t))} | f^\pi(0) = s] \tag{6}$$

The optimal state value function $V^*$ is defined by

$$V_s^* = \sup_\pi V_s^\pi, \quad s \in S \tag{7}$$

The Markov decision problem is to evaluate the function $V^*$. Once this is done, an optimal policy is easily determined.

Markov decision problems are easiest when the discount $\gamma$ is strictly smaller than 1. For the undiscounted case ($\gamma = 1$), we will assume throughout that there is a reward-free state, say state 1, which is absorbing; that is, $p_{11}(a) = 1$ and $r_{1u} = 0$ for all $a \in \mathcal{A}$. The objective is then to reach that state at maximum expected reward. We say that a stationary policy is proper if the probability of being at the absorbing state converges to 1 as time converges to infinity; otherwise, we say that the policy is improper.

We define the dynamic programming operator $T : \mathbb{R}^{|\mathcal{S}|} \mapsto \mathbb{R}^{|\mathcal{S}|}$, with components $T_i$, by letting

$$T_s(V) = \max_{a \in \mathcal{A}} \{E[r_{sa}] + \gamma \sum_{s' \in \mathcal{S}} p_{ss'}(a) V_{s'}\} \tag{8}$$

It is well known that if $\gamma < 1$, then $T$ is a contraction with respect to the norm $\| \cdot \|_\infty$ and $V^*$ is its unique fixed point.

For Generalized Q-learning algorithm, assume that there are $N$ estimators of action-values $Q^1, \ldots, Q^N$. Let $m$ be the cardinality of $\mathcal{S}$ and $n$ be the cardinality of $\mathcal{A}$. We use a discrete index variable $t$ in order to count iterations. Denote $Q^{ij}(t) = Q^i(t+j)$. After $t$ iterations, we have a vector $Q(t) \in \mathbb{R}^w$ and $w = mnNK$, with components $Q_{sa}^{ij}(t)$, $(s, a) \in \mathcal{S} \times \mathcal{A}$, $i \in \{1, \ldots, N\}$, and $j \in \{0, \ldots, K-1\}$.

By definition, for $j \in \{1, \ldots, K-1\}$, we have

$$Q_{sa}^{ij}(t+1) = Q_{sa}^{i,j-1}(t). \tag{9}$$

For $j = 0$, we have $Q_{sa}^{i0} = Q_{sa}^i$. And we update according to the formula

$$Q_{sa}^i(t+1) = Q_{sa}^i(t) + \alpha_{sa}^i(t)[Y^{GQ}(t) - Q_{sa}^i(t)] \tag{10}$$

where

$$Y^{GQ}(t) = r_{sa} + \gamma Q_{f(s,a)}^{GQ}(t). \tag{11}$$

Here, each $\alpha_{sa}^i(t)$ is a nonnegative step-size coefficient which is set to zero for those $(s, a) \in \mathcal{S} \times \mathcal{A}$ and $i \in \{1, \ldots, N\}$ for which $Q_{sa}^i$ is not to be updated at the current iteration. Furthermore, $r_{sa}$ is a random sample of the immediate reward if action $a$ is applied at state $s$. $f(s, a)$ is a random successor state which is equal to $s'$ with probability $p_{ss'}(a)$. Finally, $Q_s^{GQ}(t)$ is defined as

$$Q_s^{GQ}(t) = G(Q_s(t)) \tag{12}$$

where $G$ is a mapping from $\mathbb{R}^{nNK}$ to $\mathbb{R}$. It is understood that all random samples that are drawn in the course of the algorithm are drawn independently.

Since for $j \in \{1, \ldots, K-1\}$, we just preserve current available action-values, we only focus on the case that $j = 0$ in the sequel. Let $F$ be the mapping from $\mathbb{R}^{mnNK}$ into $\mathbb{R}^{mnN}$ with components $F_{sa}^i$ defined by

$$F_{sa}^i(Q) = E[r_{sa}] + \gamma E[Q_{f(s,a)}^{GQ}] \tag{13}$$

and note that

$$E[Q_s^{GQ}] = \sum_{s' \in \mathcal{S}} p_{ss'}(a) Q_{s'}^{GQ} \tag{14}$$

If $F_{sa}^i(Q(t)) = Q(t)_{sa}^i$, we can do $K$ more updates such that $Q(t)_a^{ij} = Q(t)_a^{kl}, \forall i, k \in \{1, \dots, N\}$, $\forall j, l \in \{0, \dots, K-1\}$, and $\forall a \in \mathcal{A}$.

In view of Equation 13, Equation 10 can be written as

$$Q_{sa}^i(t+1) = Q_{sa}^i(t) + \alpha_{sa}^i(t)[F_{sa}^i(Q(t)) - Q_{sa}^i(t) + w_{sa}^i(t)] \tag{15}$$

where

$$w_{sa}^i(t) = r_{sa} - E[r_{sa}] + \gamma(Q_{f(s,a)}^{GQ}(t) - E[Q_{f(s,a)}^{GQ}(t)|\mathcal{F}(t)]) \tag{16}$$

and $\mathcal{F}(t)$ represents the history of the algorithm during the first $t$ iterations. The expectation in the expression $E[Q_{f(s,a)}^{GQ}(t)|\mathcal{F}(t)]$ is with respect to $f(s,a)$.

## B.2 KEY LEMMAS AND THE PROOFS

**Lemma 2** *Assume Assumption 1 holds for function $G$ in Generalized Q-learning. Then we have*

$$E[w_{sa}^2(t)|\mathcal{F}(t)] \le Var(r_{sa}) + \max_{i \in \{1,\dots,N\}} \max_{\tau \le t} \max_{(s,a) \in \mathcal{S} \times \mathcal{A}} |Q_{sa}^i(\tau)|^2.$$

**Proof.** Under Assumption 1, the conditional variance of $Q_{f(s,a)}^{GQ}$ given $\mathcal{F}(t)$, is bounded above by the largest possible value that this random variable could take, which is $\max_{i \in \{1,\dots,N\}} \max_{j \in \{0,\dots,K-1\}} \max_{(s,a) \in \mathcal{S} \times \mathcal{A}} |Q_{sa}^i(t-j)|^2$. We then take the conditional variance of both sides of Equation 16, to obtain

$$E[w_{sa}^2(t)|\mathcal{F}(t)] \le Var(r_{sa}) + \max_{i \in \{1,\dots,N\}} \max_{\tau \le t} \max_{(s,a) \in \mathcal{S} \times \mathcal{A}} |Q_{sa}^i(\tau)|^2 \tag{17}$$

We have assumed here that $r_{sa}$ is independent from $f(s,a)$. If it is not, the right-hand side in the last inequality must be multiplied by 2, but the conclusion does not change. ■

**Lemma 3** *$F$ is a contraction mapping, in each of the following cases:*

*(i) $\gamma < 1$.*

*(ii) $\gamma = 1$ and $\forall a \in \mathcal{A}, Q_{s_1a}^i(t=0) = 0$ where $s_1$ is an absorbing state. All policies are proper.*

**Proof.** For discounted problems ($\gamma < 1$), Equation 13 easily yields $\forall Q, Q'$,

$$|F_{sa}^i(Q) - F_{sa}^i(Q')| \le \gamma \max_{s \in \mathcal{S}} |Q_s^{GQ} - Q_s'^{GQ}| \tag{18}$$

In particular, $F$ is a contraction mapping, with respect to the maximum norm $\|\cdot\|_\infty$.

For undiscounted problems ($\gamma = 1$), our assumptions on the absorbing state $s_1$ imply that the update equation for $Q_{s_1a}^i$ degenerates to $Q_{s_1a}^i(t+1) = Q_{s_1a}^i(t)$, for all $t$. We will be assuming in the sequel, that $Q_{s_1a}^i$ is initialized at zero. This leads to an equivalent description of the algorithm in which the mappings $F_{sa}^i$ of Equation 13 are replaced by mappings $\tilde{F}_{sa}^i$ satisfying $\tilde{F}_{sa}^i = F_{sa}^i$ if $s \ne s_1$ and $\tilde{F}_{s_1a}^i(Q) = 0$ for all $a \in \mathcal{A}$, $i \in \{1, \dots, N\}$ and $Q \in \mathbb{R}^n$.

Let us consider the special case where every policy is proper. By Proposition 2.2 in the work of (Bertsekas & Tsitsiklis, 1996), there exists a vector $v > 0$ such that $T$ is a contraction with respect to the norm $\|\cdot\|_v$. In fact, a close examination of the proof of this Proposition 2.2 shows that this proof is easily extended to show that the mapping $\tilde{F}$ (with components $\tilde{F}_{sa}^i$) is a contraction with respect to the norm $\|\cdot\|_z$, where $z_{sa}^i = v_s$ for every $a \in \mathcal{A}$ and $i \in \{1, \dots, N\}$. ■

### B.3 MODELS AND ASSUMPTIONS

In this section, we describe the algorithmic model to be employed and state some assumptions that will be imposed.

The algorithm consists of noisy updates of a vector $x \in \mathbb{R}^n$, for the purpose of solving a system of equations of the form $F(x) = x$. Here $F$ is assumed to be a mapping from $\mathbb{R}^n$ into itself. Let $F_1, \ldots, F_n \colon \mathbb{R}^n \mapsto \mathbb{R}$ be the corresponding component mappings; that is, $F(x) = (F_1(x), \ldots, F_n(x))$ for all $x \in \mathbb{R}^n$.

Let $\mathcal{N}$ be the set of non-negative integers. We employ a discrete "time" variable $t$, taking values in $\mathcal{N}$. This variable need not have any relation with real time; rather, it is used to index successive updates. Let $x(t)$ be the value of the vector $x$ at time $t$ and let $x_i(t)$ denote its $i$th component. Let $T^i$ be an infinite subset of $\mathcal{N}$ indicating the set of times at which an update of $x_i$ is performed. We assume that

$$x_i(t+1) = x_i(t), \quad t \notin T^i \tag{19}$$

Regarding the times that $x_i$ is updated, we postulate an update equation of the form

$$x_i(t+1) = x_i(t) + \alpha_i(t)(F_i(x^i(t)) - x_i(t) + w_i(t)), \quad t \in T^i \tag{20}$$

Here, $\alpha(t)$ is a step-size parameter belonging to $[0, 1]$, $w_i(t)$ is a noise term, and $x_i(t)$ is a vector of possibly outdated components of $x$. In particular, we assume that

$$x^i(t) = (x_1(\tau_1^i(t)), \ldots, x_n(\tau_n^i(t))), \quad t \in T^i \tag{21}$$

where each $\tau_j^i(t)$ is an integer satisfying $0 \le \tau_j^i(t) \le t$. If no information is outdated, we have $\tau_j^i(t) = t$ and $x^i(t) = x(t)$ for all $t$; the reader may wish to think primarily of this case. For an interpretation of the general case, see (Bertsekas & Tsitsiklis, 1989). In order to bring Eqs. 19 and 20 into a unified form, it is convenient to assume that $\alpha_i(t), w_i(t)$, and $\tau_j^i(t)$ are defined for every $i$, $j$, and $t$, but that $\alpha_i(t) = 0$ and $\tau_j^i(t) = t$ for $t \notin T^i$.

We will now continue with our assumptions. All variables introduced so far $(x(t), \tau_j^i(t), \alpha_i(t), w_i(t))$ are viewed as random variables defined on a probability space $(\Omega, \mathcal{F}, \mathcal{P})$ and the assumptions deal primarily with the dependencies between these random variables. Our assumptions also involve an increasing sequence $\{\mathcal{F}(t)\}_{t=0}^{\infty}$ of subfields of $\mathcal{F}$. Intuitively, $\mathcal{F}(t)$ is meant to represent the history of the algorithm up to, and including the point at which the step-sizes $\alpha_i(t)$ for the $t$th iteration are selected, but just before the noise term $w_i(t)$ is generated. Also, the measure-theoretic terminology that "a random variable $Z$ is $\mathcal{F}(t)$-measurable" has the intuitive meaning that $Z$ is completely determined by the history represented by $\mathcal{F}(t)$.

The first assumption, which is the same as the total asynchronism assumption of Bertsekas & Tsitsiklis (1989), guarantees that even though information can be outdated, any old information is eventually discarded.

**Assumption 3** *For any $i$ and $j$, $\lim_{t \to \infty} \tau_j^i(t) = \infty$, with probability 1.*

Our next assumption refers to the statistics of the random variables involved in the algorithm.

**Assumption 4** *Let $\{\mathcal{F}(t)\}_{t=0}^{\infty}$ be an increasing sequence of subfields of $\mathcal{F}$.*

  (i) *$x(0)$ is $\mathcal{F}(0)$-measurable.*

 (ii) *For every $i$ and $t$, $w_i(t)$ is $\mathcal{F}(t+1)$-measurable.*

(iii) *For every $i$, $j$ and $t$, $\alpha_i(t)$ and $\tau_j^i(t)$ are $\mathcal{F}(t)$-measurable.*

(iv) *For every $i$ and $t$, we have $E[w_i(t)|\mathcal{F}(t)] = 0$.*

 (v) *There exist (deterministic) constants $A$ and $B$ such that*

$$E[w_i^2(t)|\mathcal{F}(t)] \le A + B \max_j \max_{\tau \le t} |x_j(\tau)|^2, \quad \forall i, t \tag{22}$$

Assumption 4 allows for the possibility of deciding whether to update a particular component $x_i$ at time $t$, based on the past history of the process. In this case, the step-size $\alpha_i(t)$ becomes a random variable. However, part $(iii)$ of the assumption requires that the choice of the components to be updated must be made without anticipatory knowledge of the noise variables $w_i$ that have not yet been realized.

Finally, we introduce a few alternative assumptions on the structure of the iteration mapping $F$. We first need some notation: if $x, y \in \mathbb{R}^n$, the inequality $x \leq y$ is to be interpreted as $x_i \leq y_i$ for all $i$. Furthermore, for any positive vector $v = (v_1, \ldots, v_n)$, we define a norm $\| \cdot \|_v$ on $\mathbb{R}^n$ by letting

$$\|x\|_v = \max_i \frac{|x_i|}{v_i}, \quad x \in \mathbb{R}^n \tag{23}$$

Notice that in the special case where all components of $v$ are equal to 1, $\| \cdot \|_v$ is the same as the maximum norm $\| \cdot \|_\infty$.

**Assumption 5** *Let $F : \mathbb{R}^n \mapsto \mathbb{R}^n$.*

*(i) The mapping $F$ is monotone; that is, if $x \leq y$, then $F(x) \leq F(y)$.*

*(ii) The mapping $F$ is continuous.*

*(iii) The mapping $F$ has a unique fixed point $x^*$.*

*(iv) If $e \in \mathbb{R}^n$ is the vector with all components equal to 1, and $r$ is a positive scalar, then*

$$F(x) - re \leq F(x - re) \leq F(x + re) \leq F(x) + re \tag{24}$$

**Assumption 6** *There exists a vector $x^* \in \mathbb{R}^n$, a positive vector $v$, and a scalar $\beta \in [0, 1)$, such that*

$$\|F(x) - x^*\|_v \leq \beta \|x - x^*\|_v, \quad \forall x \in \mathbb{R}^n \tag{25}$$

**Assumption 7** *There exists a positive vector $v$, a scalar $\beta \in [0, 1)$, and a scalar $D$ such that*

$$\|F(x)\|_v \leq \beta \|x\|_v + D, \quad \forall x \in \mathbb{R}^n \tag{26}$$

**Assumption 8** *There exists at least one proper stationary policy. Every improper stationary policy yields infinite expected cost for at least one initial state.*

**Theorem 3** *Let Assumptions 3, 4, 2, and 7 hold. Then the sequence $x(t)$ is bounded with probability 1.*

**Theorem 4** *Let Assumptions 3, 4, 2, and 5 hold. Furthermore, suppose that $x(t)$ is bounded with probability 1. Then $x(t)$ converges to $x^*$ with probability 1.*

**Theorem 5** *Let Assumptions 3, 4, 2, and 6 hold. Then $x(t)$ converges to $x^*$ with probability 1.*

Detailed proofs of Theorems 3, 4, and 5 can be found in the work of Bertsekas & Tsitsiklis (1989).

### B.4 PROOF OF THEOREM 2

We first state Theorem 2 here again and then show the proof.

**Theorem 2** Assume a finite MDP $(\mathcal{S}, \mathcal{A}, \mathrm{P}, R)$ and that Assumption 1 and 2 hold. Then the action-value functions in Generalized Q-learning, using tabular update in Equation (3), will converge to the optimal action-value function with probability 1, in each of the following cases:

(i) $\gamma < 1$.

(ii) $\gamma = 1$ and $\forall a \in \mathcal{A}, Q_{s_1 a}^i(t = 0) = 0$ where $s_1$ is an absorbing state. All policies are proper.

**Proof.** We first check Assumptions 3, 4, 2, and 6 in Section B.3 are satisfied. Then we simply apply Theorem 5 to Generalized Q-learning.

Assumption 3 is satisfied in the special case where $\tau_j^i(t) = t$, which is what was implicitly assumed in Equation 10, but can be also satisfied even if we allow for outdated information.

Regarding Assumption 4, parts $(i)$ and $(ii)$ of the assumption are then automatically valid. Part $(iii)$ is quite natural: in particular, it assumes that the required samples are generated after we decide which components to update during the current iteration. Part $(iv)$ is automatic from Equation 16. Part $(v)$ is satisfied by Lemma 2.

Assumption 2 needs to be imposed on the step-sizes employed by the Generalized Q-learning algorithm. This assumption is standard for stochastic approximation algorithms. In particular, it requires that every state-action pair $(s, a)$ is simulated an infinite number of times.

By Lemma 3, $F$ is a contraction mapping. Assumption 6 is satisfied.

All assumptions required by Theorem 5 are verified, convergence then follows from Theorem 5.

∎

## C  ADDITIONAL EMPIRICAL RESULTS

### C.1  MDP RESULTS

Comparison of three algorithms using the simple MDP in Figure 1 with different values of $\mu$ is shown in Figure 6. For $\mu = +0.1$, the learning curves of action value $Q(A, \text{Left})$ are shown in $(a)$. Here, the true action value $Q(A, \text{Left})$ is $+0.1$. For $\mu = -0.1$, the learning curves of action value $Q(A, \text{Left})$ are shown in $(b)$. The true action value $Q(A, \text{Left})$ is $-0.1$. All results were averaged over $5,000$ runs.

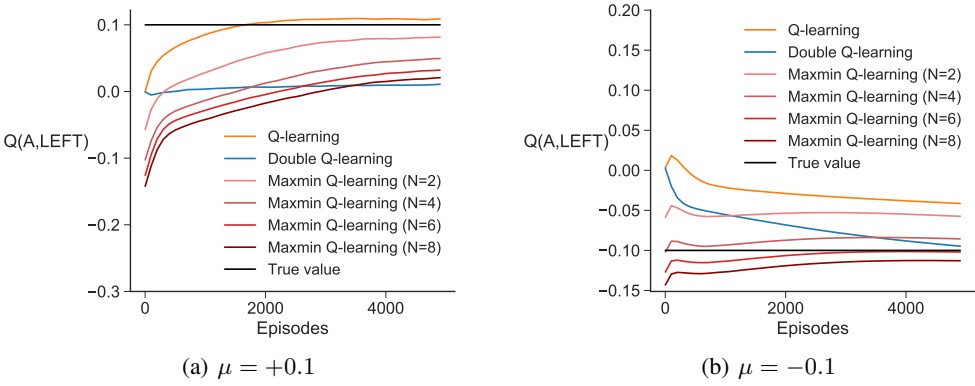

(a) $\mu = +0.1$        (b) $\mu = -0.1$

Figure 6: MDP results

### C.2  MOUNTAIN CAR RESULTS

Comparison of four algorithms on Mountain Car under different reward settings is shown in Figure 7. All experimental results were averaged over $100$ runs. Note that for reward variance $\sigma^2 = 50$, both Q-learning and Averaged Q-learning fail to reach the goal position in $5,000$ steps so there are no learning curves shown in Figure 7 $(d)$ for these two algorithms.

### C.3  BENCHMARK ENVIRONMENT RESULTS

The sensitivity analysis results of seven benchmark environment are shown in Figure 8.

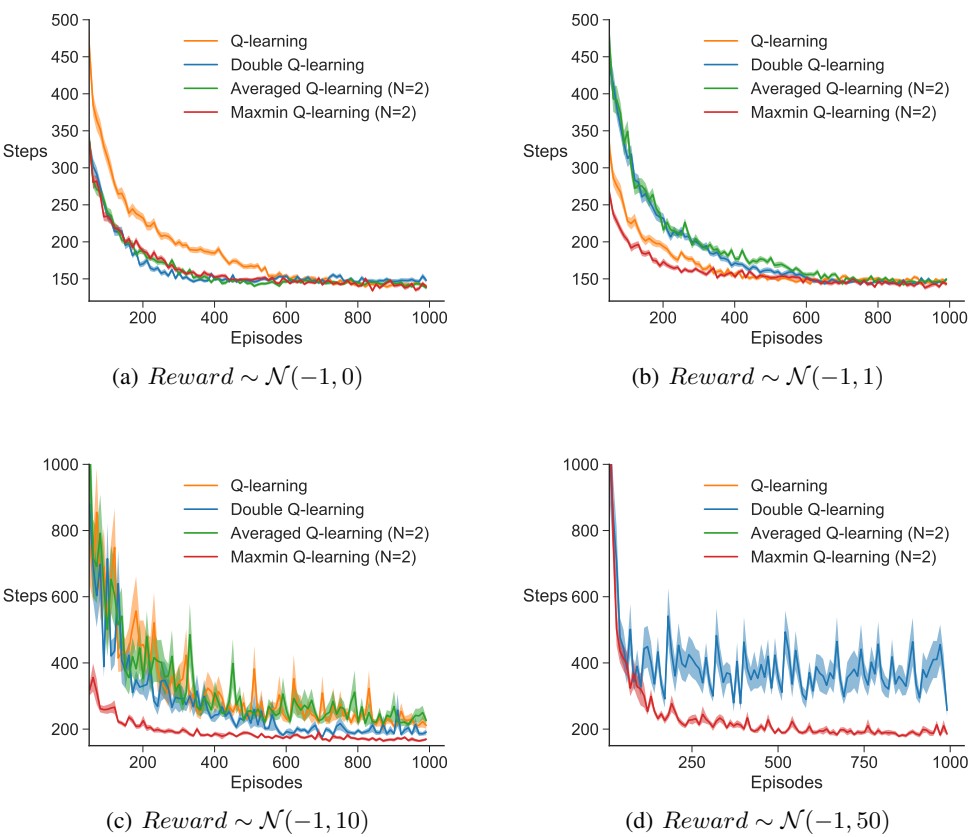

Figure 7: Mountain Car results

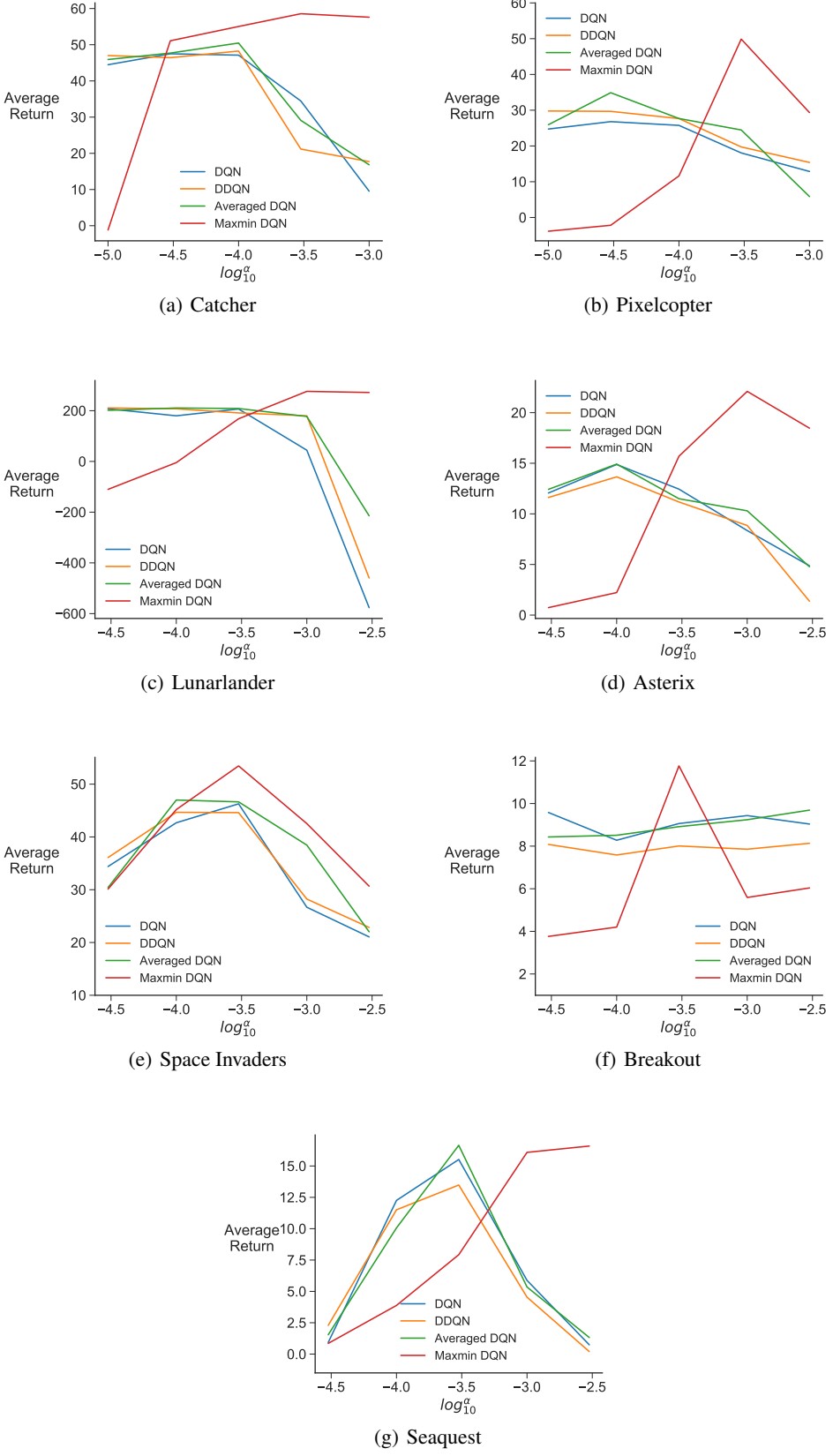

Figure 8: Sensitivity analysis

