# OpenReview forum: "Maxmin Q-learning: Controlling the Estimation Bias of Q-learning"
_ICLR.cc/2020/Conference — Accept (Poster)_

### Official Review · AnonReviewer1 · 2019-10-14
**Official Blind Review #1**

**Rating:** 3

**Review:**

The paper tackles the problem of bias in target Q-values when performing Q-learning.  The paper proposes a technique for computing target Q-values, by first taking the min over an ensemble of learned Q-values and then taking the max over actions.  The paper provides some theoretical properties of this technique: (1) the bias of the estimator can be somewhat controlled by the size of the ensemble; (2) performing Q-learning with these target values is convergent. Experimental results show that the proposed technique can provide performance improvement on a number of tasks.

Overall, this paper is a modest contribution to the field, since variants of this technique are known and the theoretical arguments are derivatives of known arguments, which places it roughly borderline for an ICLR conference paper.

My comments:
-- The paper is very well-written. Thank you for putting the effort to provide clear writing.
-- The idea of computing a target value as the minimum of an ensemble is well-known in continuous control.  See https://arxiv.org/abs/1802.09477 as well as a number of works which follow it.
-- The method is motivated as a way to control over/under-estimation.  However, the theoretical arguments show that this depends on N, M, and tau (unknown). Are there any ways to choose N other than hyperparameter tuning?

**Experience Assessment:**

I have published in this field for several years.

**Review Assessment: Checking Correctness Of Derivations And Theory:**

I assessed the sensibility of the derivations and theory.

**Review Assessment: Checking Correctness Of Experiments:**

I assessed the sensibility of the experiments.

**Review Assessment: Thoroughness In Paper Reading:**

I read the paper at least twice and used my best judgement in assessing the paper.

---

> ### Author Response · Authors · 2019-11-09
> **Reply to Review #1**
>
> Thank you for your valuable comments. We will address your concerns point by point.
>
> First, thank you for pointing us to this control work, which we were not aware of. However, we would like to point out the significant difference between Maxmin Q-learning and TD3 in the paper you mentioned when computing the target value. For Maxmin Q-learning, we first take the minimum of action-values among all N estimators and then choose the action that maximizes these minimum action-values, i.e. $Y^{MQ} = r + \gamma \max_{a \in A} \min_{i=1,\cdots,N} Q^{i}(s,a)$. However, in TD3, the action is chosen first by some policy $\pi$, and then the minimum action-value is selected as the target value, i.e. $Y^{TD3} = r + \gamma \min_{i=1,2} Q^{i}(s,\pi(s))$. The policy $\pi$ is expected to converge to the optimal policy, so $Q^{i}(s,\pi(s)) \approx \max_{a \in A} Q^{i}(s,a)$. Thus, we have $Y^{TD3} \approx r + \gamma \min_{i=1,2} \max_{a \in A} Q^{i}(s,a)$. Note that the order of taking minimum and maximum is different. By the max–min inequality, it is easy to get $Y^{TD3} > Y^{MQ}$. Furthermore, using the same method as in our paper, we can get $E[Y^{TD3}] > E[Y^{True}]$ where $Y^{True} = r + \gamma \max_{a \in A} Q^{*}(s,a)$. Thus TD3 still suffers from the overestimation bias, while we can adjust $N$ in Maxmin Q-learning to reduce the bias from positive to negative. In conclusion, these two techniques may seem to be similar at first glance, but they are actually quite different and lead to different properties. We will cite the TD3 paper and add one more paragraph to discuss the difference in our revised paper.
>
> Also, there is no theoretical analysis for applying the minimum operator in TD3 paper, whereas we present a theoretical analysis not only for bias control but also for variance reduction. Actually, when we first tried to solve the overestimation bias problem, we also considered a similar approach to TD3 (we called it Minmax Q-learning as a counterpart). However, after some derivatives and analyses, we found that Maxmin Q-learning would be better than Minmax Q-learning theoretically, in terms of reducing overestimation bias. These theoretical analyses guided our design of the Maxmin Q-learning algorithm. They are important and non-trivial. You are right that "the theoretical arguments are derivatives of known arguments", but this is the case for many theoretical arguments. We have not come up with a new proof technique, but do have a novel theoretical result characterizing a new algorithm.
>
> Choosing the optimal N is not straightforward. It is a parameter that can be tuned, to specialize for each different problem setting. The theory does provide some guidance, namely by setting $t_{MN}=1/2$ to remove bias. For example, when M=4, a choice of N=3 reduces the bias to near 0 (more results are shown in Figure 5). This may not be the right choice, though, if the noise does not satisfy the assumptions. In our experiments, we found that the optimal N was usually between 2 to 9, and that performance was usually improved with relatively small N. You are correct that future work should investigate how best to select N.

---

> > ### Comment · AnonReviewer1 · 2019-11-13
> > **Comment**
> >
> > The distinction between min-max and max-min depends on implementation. I believe most implementations correspond to the max-min (as in your proposed algorithm).
> >
> > For example, you can see in SAC here: https://github.com/rail-berkeley/softlearning/blob/master/softlearning/algorithms/sac.py#L238
> > The policy is learned to maximize the minimum of a Q-ensemble. Thus, the backup Q-value would be max-min of the Q-ensemble.

---

> > > ### Author Response · Authors · 2019-11-14
> > > **Further clarifications**
> > >
> > > Thank you for your comments and referring us to the implementation of SAC.
> > >
> > > However, we would like to point out that Maxmin Q-learning and SAC/TD3 differ, and the implementation choice of max-min or min-max is significant. The ordering produces algorithms that are fundamentally different.
> > >
> > > In the implementation of SAC, for both the actor update and critic update, the actions are selected first by some policy (
> > > https://github.com/rail-berkeley/softlearning/blob/master/softlearning/algorithms/sac.py#L115, https://github.com/rail-berkeley/softlearning/blob/master/softlearning/algorithms/sac.py#L192), then the minimum action-values are taken(https://github.com/rail-berkeley/softlearning/blob/master/softlearning/algorithms/sac.py#L126, https://github.com/rail-berkeley/softlearning/blob/master/softlearning/algorithms/sac.py#L233). Ignoring the entropy term, the target action-value should be $Y^{SAC} \approx r + \gamma \min_{i=1,2} \max_{a \in A} Q^{i}(s,a)$ as we explained previously. But for Maxmin Q-learning, the target action-value is $Y^{MQ} = r + \gamma \max_{a \in A} \min_{i=1,\cdots,N} Q^{i}(s,a)$. Notice that the order of taking the minimum and maximum are different in SAC and Maxmin Q-learning. We previously explained why $Y^{SAC}$ still suffers from overestimation bias, and that $Y^{MQ}$ is better than $Y^{SAC}$.
> > >
> > > We agree that "the policy is learned to maximize the minimum of a Q-ensemble". But this Q-ensemble differs from Maxmin Q-learning. Let $\theta$ be the weights for the actor. The minimum of the Q-ensemble you mentioned here is actually $\min_{i=1,2} Q^{i}(s,\pi_{\theta}(s))$. Ignoring other terms in policy_kl_losses, we write down the formula: $\max_{\theta} \min_{i=1,2} Q^{i}(s,\pi_{\theta}(s))$. The policy $\pi_{\theta}$ is expected to converge to the optimal policy, so $Q^{i}(s,\pi_{\theta}(s)) \approx \max_{a \in A} Q^{i}(s,a)$. Thus, what the policy really learns is to do $\max_{\theta} \min_{i=1,2} \max_{a \in A} Q^{i}(s,a)$. Note that the first maximum operator is applied on $\theta$ and this is different from the $\max_{a \in A} \min_{i=1,\cdots,N} Q^{i}(s,a)$ in Maxmin Q-learning.
> > >
> > > Both SAC and Maxmin Q-learning apply a minimum operator. However, where and how they apply it is different. And there is a theoretical reason for the approach we use in Maxmin Q-learning. Again, we will cite the SAC paper and update our paper in the future.

---

> > > > ### Comment · AnonReviewer1 · 2019-11-14
> > > > **Response**
> > > >
> > > > Thanks for following up.  There seems to be some confusion remaining which I will try to resolve.
> > > >
> > > > The target values in SAC are computed as \min_{i} Q^i(s, \pi(s)).
> > > >
> > > > What is \pi(s)? The policy is trained to optimize \max_\pi\min_i Q^i(s, \pi(s)), ignoring KL terms.  Ignoring issues of expressibility, the optimal \pi(s) is \pi(s) = \argmax_a \min_i Q^i(s, a).
> > > >
> > > > Going back to the target value, we can now see that the target value at s would be \min_{i} Q^i(s, \argmax_a \min_i Q^i(s, a)) = \max_a \min_{i} Q^i(s, a), which is exactly your proposed max-min Q learning.

---

> > > > > ### Author Response · Authors · 2019-11-14
> > > > > **Further response**
> > > > >
> > > > > Thank you for your great clarifications! Your derivations are definitely correct based on your assumptions. However, we still want to point out some remaining issues.
> > > > >
> > > > > First, there is no theoretical guarantee that the learned policy for SAC will converge to the optimal policy due to the issues of expressibility, or just because the optimizer is stuck in some local minimums. Also, there is no convergence proof in the SAC paper for the case of applying two or more Q functions with the max-min operator. What's more, during the training stage when the policy has not reached to the optimal policy, the target action-values in Maxmin Q-learning and SAC are also different. In all these cases, $\pi(s) \neq \arg \max_a \min_i Q^i(s, a)$ and $\min_{i} Q^i(s, \arg \max_a \min_i Q^i(s, a)) \neq \max_a \min_{i} Q^i(s, a)$. However, Maxmin Q-learning has the same target action-value $\max_a \min_{i} Q^i(s, a)$ no matter whether the policy converges to the optimal policy or not. This is one advantage of Maxmin Q-learning compared to SAC.
> > > > >
> > > > > Second, we would like to further emphasize that SAC and Maxmin Q-learning are fundamentally different algorithms in nature. SAC is an algorithm sitting between policy-based and value-based methods while Maxmin Q-learning is a value-based method. Even if the learned policy converges to the optimal policy, the target action-values of Maxmin Q-learning and SAC before convergence are not the same. These two algorithms take different ways to converge and this may influence the behavior greatly. For example, both Q-learning and policy gradient methods try to maximize the total return, yet they behave differently.
> > > > >
> > > > > Some techniques are effective in practice. However, we are lack of a rigorous theoretical analysis of why these techniques work and when they work. For example, there is no theoretical analysis in either the TD3 paper or the SAC paper for the effect of applying the minimum operator. There is also no theoretical guide for choosing $N$ such that the overestimation bias is reduced to near 0. But we do provide non-trivial theoretical results in our paper. This is also an important contribution of our work.
> > > > >
> > > > > In some sense, most algorithms can be thought of as some modifications of existing ones, but we cannot deny that they have their own values.
> > > > >
> > > > > Besides, we also propose a novel Generalized Q-learning framework and prove the convergence of Generalized Q-learning under reasonable assumptions. The convergence proofs of many algorithms are now contained in one framework, such as Q-learning, Maxmin Q-learning, Ensemble Q-learning, Averaged Q-learning, and Historical Best Q-learning. This Generalized Q-learning framework is a handy tool to analyze convergence properties of many Q-learning variants and may even inspire researchers to invent better algorithms.

---

### Official Review · AnonReviewer3 · 2019-10-22
**Official Blind Review #3**

**Rating:** 6

**Review:**

This paper proposes a new Q learning algorithm framework: maxmin Q-learning, to address the overestimation bias issue of Q learning. The main contributions of this paper are three folds: 1) It provides an inspiring example on overestimation/underestimation of Q learning. 2) Generalize Q learning by a new maxmin Q-learning by maintaining independent Q estimator and interact them in a max-min way for the update. 3) Provide both theoretical and empirical analyses of their algorithm.

I have two main concerns for this paper:
1) When is your algorithm useful? What's your criterion of picking the hyper-parameters (e.g. number of Q functions you want to learn).
2) Comparison to more intriguing way for jointly update of multiple Q functions, like soft Q learning.

For the first concern, the paper has shown an interesting example in Figure 2. However, it seems that we cannot decide whether overestimation or underestimation will help the exploration, since the reward function is often unknown in real world. And in both cases, maxmin Q learning is not the best algorithm than either Q learning and double q learning. On the other hand, if we use a softmax policy for Q function, e.g. $\pi(a|s) \propto \exp(\alpha Q(s,a))$, a drift for Q learning(e.g. Q(s,a) = Q*(s,a) + c) has no effect on our policy. I believe in this case we should more focusing on the inner difference between different value of Q function, rather than comparing our estimate Q function with the true Q*.

For the second concern, we can view the framework of maxmin q learning as a joint update scheme for different Q function. In experimental part, the comparison is not fair since the paper use multiple Q function to compare with single or double Q function. One reasonable baseline is to update N different Q function, and take the minimum of the final Q function as our decision policy, compare with maxmin Q learning with N different Q function. Another baseline the paper should consider is soft Q learning, where it maintain multiple Q function and jointly update Q different function to maximize the entropy while moving towards an improvement Q.

Overall, I believe the idea of the paper is novel and interesting, but further improvements should be added in order to improve the score the paper.

**Experience Assessment:**

I have read many papers in this area.

**Review Assessment: Checking Correctness Of Derivations And Theory:**

I carefully checked the derivations and theory.

**Review Assessment: Checking Correctness Of Experiments:**

I carefully checked the experiments.

**Review Assessment: Thoroughness In Paper Reading:**

I read the paper thoroughly.

---

> ### Author Response · Authors · 2019-11-09
> **Reply to Reivew #3**
>
> We appreciate your feedback.
>
> For the first concern, you are right, we cannot know for an unknown environment whether overestimation or underestimation will help. This is exactly what we show in section 3 -- the optimal bias is environment dependent. So the optimal N is also environment dependent. N is a parameter that can be tuned, to specialize to each different problem. However, the theory does provide some guidance, namely by setting $t_{MN}=1/2$ to remove bias. For example, when M=4, a choice of N=3 reduces the bias to near 0 (more results are shown in Figure 5). This may not be the right choice, though, if the noise does not satisfy the assumptions. In our experiments, we found that the optimal N was usually between 2 to 9, and that performance was usually improved with relatively small N. But, more work is needed to better understand more generally how to select N automatically.
>
> The MDP we present in section 3 is designed as a motivating example to show that overestimation and underestimation bias can both help and hurt. Note that we fixed the step-size and other hyperparameters for all algorithms (instead of tuning them in order to achieve the best performance) since this MDP was not used as a benchmark for performance comparison. It is just an illustrative example.
>
> We are not exactly sure what you mean by your comment that "a drift for Q learning (e.g. $Q(s,a) = Q^{*}(s,a) + c$) has no effect on our policy". What is c? If it is a constant for all actions, absolutely it has no effect. But, the problem we are solving is that, due to stochasticity, a different constant could be added $Q(s,a)$ for each $a$. This will effect the policy. If c is random, could you clarify further what you mean here?
>
> You are right that our Maxmin Q-learning is a joint update scheme for different Q functions, and one of our contributions is that we provide a convergence proof for such a framework under reasonable assumptions. However, we politely disagree with your claim that our empirical comparison is unfair. Note that on Mountain Car (Figure 3), we compare Double Q-learning, Averaged Q-learning (N=2), and Maxmin Q-learning (N=2). Here, all three algorithms have two Q functions and Maxmin Q-learning shows significant robustness and achieves better performance. Similarly, in the other seven more complex environments, both Maxmin DQN and Averaged DQN learn multiple Q functions. And the number of Q functions is tuned in the same scope. Again, Maxmin DQN outperforms Averaged DQN. So we believe the comparison is fair.
>
> We are also unsure about what you mean by "one reasonable baseline is to update N different Q function, and take the minimum of the final Q function as our decision policy." This could mean the following. N Q functions are learned with Q-learning (rather than say with the Maxmin update). On each step, the agent selects actions using the max action from the minimum of the Q functions. (If this is not what you intended, please do clarify). Theoretically, this strategy also incurs overestimation bias since it uses the same target action-value as Q-learning to update. Further it does not reduce estimation variance because it uses only one Q function to update.
>
> For Soft Q-learning (SQL), we assume you mean the algorithm from the paper Reinforcement Learning with Deep Energy-Based Policies by Haarnoja et al., (again, please correct us if we are wrong). Maxmin Q-learning is a value-based method for discrete control to flexibly control bias and reduce variance. In contrast, SQL is a policy-based method for continuous control, with just one action-value function (the policy is a Gibbs distribution on Q function). It is not clear why we would compare to SQL, since it only uses one Q function and tackles a different problem. If you can further clarify why we should compare to SQL, we would be happy to respond further.

---

> > ### Comment · AnonReviewer3 · 2019-11-13
> > **Thank you for the response, a few more questions.**
> >
> > Thank you for your detailed response. I think I had misinterpretation on the paper at the time I wrote the rebuttal but your response clarifies most of my concerns.
> >
> > > overestimation or underestimation in section 3.
> > I think I initially misunderstand your novelty as solving the 'overestimation' of Q learning. I now understand what the framework not only solves 'overestimation' but also solves 'underestimation' (which maybe the problem brought by double q learning). Thanks for clarifying that, I think you can highlight that in your abstraction to make it clearer.
> >
> > > choice of the parameter
> > Thanks for clarifying that. So based on the theory, $N$ should be chosen w.r.t to $M$, is that correct?
> >
> > > drift for Q learning
> > I mean if $c$ is a constant, though Q is not correct, the optimal policy $\pi(s) = \arg\max_a Q(s,a)$ should remain the same, similar for a softmax policy. In this sense, overestimation or underestimation seems not hurt the learned policy too much. Could you make further clarification on that?
> >
> > > reasonable baseline
> > This may not be useful, but what I mean is separately learn $N$ Q function $Q^1,...,Q^N$, and choose the policy $\pi(s) = \arg\max_a \min_i Q^i(s,a)$. This is similar to your max min framework but the only difference is we don't jointly learn $Q^i$. What you compare is just taking 'average' of $N$ Q function which may not directly comparable.
> >
> > > soft Q learning
> > I agree it looks not the same to your methods. But I am curious in the way that can I view what you jointly updated $Q^1, Q^2,..., Q^N$ as a special joint update rule for samples of a distributional $Q$ function? Distributional way may encourage exploration and thus may help the optimization. This is why I mention soft q learning. I agree that is not directly related to your contribution(reduce the bias), but it is my first intuition when I try to understand why your method can outperform the others.

---

> > > ### Author Response · Authors · 2019-11-14
> > > **Answers to some questions**
> > >
> > > Thank you for you comments!
> > >
> > > - overestimation or underestimation in section 3: Yes, you are right, "the framework not only solves 'overestimation' but also solves 'underestimation'". We will highlight this in our abstraction to make it clearer. Thank you for your advice.
> > >
> > > - choice of the parameter: Yes, the N that reduces the bias to near 0 should be chosen w.r.t to M (the number of actions).
> > >
> > > - drift for Q learning: You are right that if c is a constant, it has no influence on the optimal policy. However, c is a random variable due to stochasticity. The stochasticity may come from many sources, such as environment dynamics and function approximation error [1,2,3]. It is a noise term that may has a negative influence on the policy. In Section 4, we assume that it is a random variable that depends on state-action pair $(s,a)$, i.e. $Q(s,a)=Q^{*}(s,a)+e(s,a)$. Then in Theorem 1, we prove that by choosing a good N, the negative influence of this noise term can be diminished. Our experiments also support the theory.
> > >
> > > - reasonable baseline: Separately learning N Q functions without jointly learning $Q^i$ still suffers from overestimation bias same as Q-learning. We don't think this is useful. Taking the average of N Q functions can directly reduce overestimation bias and variance. Maxmin Q-learning also reduces overestimation bias and variance. We want to see which algorithm is better in doing this and leads to a better performance, so we compare them. They are directly comparable in the sense of controlling overestimation bias and reducing variance.
> > >
> > > - soft Q-learning: You can view Maxmin Q-learning as a special case of Generalized Q-learning which we propose in Section 6. And so is Averaged Q-learning and Double Q-learning! If "Distributional way may encourage exploration and thus may help the optimization", Averaged Q-learning should benefit from this effect too. Yet, Maxmin Q-learning still outperforms Averaged Q-learning. So Maxmin Q-learning must have some other advantages. These advantages are better bias controlling and variance reduction. It can diminish the bad influence of the noise term and produce a better policy.
> > >
> > >
> > > Again, we thank you for your questions for giving us a chance to explain the advantages of Maxmin Q-learning. If you are satisfied with our clarifications, we encourage you to update your score accordingly.
> > >
> > >
> > > [1] Sebastian Thrun and Anton Schwartz. Issues in Using Function Approximation for Reinforcement Learning. In Fourth Connectionist Models Summer School, 1993.
> > > [2] Istv ́an Szita and Andr ́as L ̋orincz. The Many Faces of Optimism: A Unifying Approach. In International Conference on Machine learning, pp. 1048–1055. ACM, 2008.
> > > [3] Alexander L. Strehl, Lihong Li, and Michael L. Littman.  Reinforcement Learning in Finite MDPs:PAC Analysis. Journal of Machine Learning Research, 10(Nov):2413–2444, 2009.

---

> > > > ### Comment · AnonReviewer3 · 2019-11-15
> > > > **Thank you for the clarification**
> > > >
> > > > Thank you for the response. I'm satisfied with the clarification and will increase my score.

---

> > > ### Author Response · Authors · 2019-11-14
> > > **Result for "reasonable baseline" on Mountain Car**
> > >
> > > For your curiosity, we run the "reasonable baseline" as you suggest on Mountain Car with reward variance 10. We call it "Maxmin Baseline". We did the same hyper-parameter sweep as what we did in our paper. The result was averaged on 100 runs.  Here is the result: https://anonymousfiles.io/bxrzs0oR/
> > >
> > > As you can see, it does not perform well.

---

### Official Review · AnonReviewer2 · 2019-10-24
**Official Blind Review #2**

**Rating:** 8

**Review:**

This paper proposes a novel variant of Q-learning, called Maxmin Q-learning, to address the issue of overestimation bias Q-learning suffers from (variance of the reward of best action leading to overestimated reward).
The idea is to keep a number of estimators each estimated using a different sub-sample, and taking the minimum of the (maximum) reward value of each.
The paper gives theoretical analyses, in terms of the reduction in the overestimation bias, as well as the convergence of a class of generalized Q-learning methods including Maxmin Q-learning.
The experiment section presents a thorough evaluation of the proposed method, including how the obtained rewards vary as a function of the variance of the reward function and as a function of learning steps, as compared to a number of existing methods such as the Double Q-learning method and its variants.
The experimental results are quite convincing, and the theoretical analyses seem solid.
Overall this is a well balanced paper which proposes a reasonable new idea, simple but effective, backed by sound theoretical analysis and well executed experimental evaluation.

**Experience Assessment:**

I have published one or two papers in this area.

**Review Assessment: Checking Correctness Of Derivations And Theory:**

I assessed the sensibility of the derivations and theory.

**Review Assessment: Checking Correctness Of Experiments:**

I assessed the sensibility of the experiments.

**Review Assessment: Thoroughness In Paper Reading:**

I read the paper at least twice and used my best judgement in assessing the paper.

---

> ### Author Response · Authors · 2019-11-09
> **Reply to Review #2**
>
> Thank you for your positive comments!

---

### Decision · Program_Chairs · 2019-12-19

**Decision:**

Accept (Poster)

**Comment:**

The authors propose the use of an ensembling scheme to remove over-estimation bias in Q-Learning. The idea is simple but well-founded on theory and backed by experimental evidence. The authors also extensively clarified distinctions between their idea and similar ideas in the reinforcement learning literature in response to reviewer concerns.